# Influence of the Surface Chemical Composition Differences between Zirconia and Titanium with the Similar Surface Structure and Roughness on Bone Formation

**DOI:** 10.3390/nano12142478

**Published:** 2022-07-19

**Authors:** Yoshiki Tokunaga, Masatsugu Hirota, Tohru Hayakawa

**Affiliations:** 1Department of Dental Engineering, Tsurumi University School of Dental Medicine, 2-1-3 Tsurumi, Tsurumi-ku, Yokohama 230-8501, Japan; hayakawa-t@tsurumi-u.ac.jp; 2Department of Education for Dental Medicine, Tsurumi University School of Dental Medicine, 2-1-3 Tsurumi, Tsurumi-ku, Yokohama 230-8501, Japan; hirota-masatsugu@tsurumi-u.ac.jp

**Keywords:** zirconia, molecular precursor method, dental implant, osseointegration, bone-to-implant contact

## Abstract

The osseointegration of zirconia (ZrO_2_) implants is still controversial. In this study, we aimed to make clear the influence of surface chemical composition, Ti or ZrO_2_, to osseointegration. First, a roughened Ti surface was prepared with a combination of large-grit sandblasting and acid treatment. Then, we applied molecular precursor solution containing Zr complex onto roughened Ti surface and can deposit thin ZrO_2_ film onto roughened Ti surface. We can change surface chemical composition from Ti to ZrO_2_ without changing the surface structure and roughness of roughened Ti. The tetragonal Zr was uniformly present on the ZrO_2_-coated Ti surface, and the surface of the ZrO_2_-coated Ti showed a higher apparent zeta potential than Ti. Ti and ZrO_2_-coated Ti rectangular plate implant was placed into the femur bone defect. After 2 and 4 weeks of implantation, histomorphometric observation revealed that the bone-to-implant contact ratio and the bone mass values for ZrO_2_-coated Ti implants inserted into the femur bone defects of the rats at 2 weeks were significantly higher than those for Ti implants (*p* < 0.05). It revealed that ZrO_2_ with a similar surface structure and roughness as that of roughened Ti promoted osteogenesis equivalent to or better than that of Ti in the early bone formation stage.

## 1. Introduction

Using titanium (Ti) as a dental implant material has advantages owing to its excellent mechanical properties, biocompatibility, and tight bonding to the bone tissue, which is known as osseointegration. However, aesthetic, and hypersensitive reactions including allergies to Ti implants have been reported [1,2,3]. An yttria-stabilized tetragonal zirconia polycrystal (Y-TZP) was used as another option as a dental implant material [4]. However, the osseointegration of zirconia (ZrO_2_) implants is controversial. Kohal et al. [5]. evaluated 65 one-piece ZrO_2_ implants incorporated in humans for three years post-operation and reported that the survival rates of ZrO_2_ implants were inferior to those of Ti implants. In contrast, Roehling et al. [6]. studied 2758 publications and reported that ZrO_2_ and Ti implants showed similar soft and hard tissue integration capacities.

It is well known that surface structure and surface roughness is key to the osseointegration of dental implants [7]. The surface treatment of Ti implants by combining the large-grit and acid-etched procedures, which is generally named SLA, produced positive effects on the activation of blood platelets and cell migration, and clinical success rate [8,9,10,11]. Surface modifications of Y-TZP implants have been reported [12,13,14], and large-grit and acid etching procedures were also applied to Y-TZP implants [12,13,14,15]. However, Y-TZP is known to exhibit high mechanical strength and chemical stability [16], and achieving a similar surface structure and roughness on Y-TZP as that on Ti is difficult. Surface roughness is a popularly used parameter for evaluating the material’s surface. Bormann et al. [17]. compared the bone tissue response to those of the micro-structured ZrO_2_ and SLA-Ti implants. ZrO_2_ implants were chemically treated with a hot solution of hydrofluoric acid. They reported that the mean roughness of the SLA-Ti implants (Sa = 1.19 μm) was approximately twice that of similar ZrO_2_ implants (Sa = 0.63 μm).

Apart from surface structure and roughness, the surface chemical composition also affects the osseointegration of dental implants. It is not clear which factor, surface structure, roughness, or chemical composition will contribute more to osseointegration for ZrO_2_ implants. It is presumed that this is the reason for the controversy regarding the osseointegration of ZrO_2_. To elucidate the osseointegration of ZrO_2_ implants, it is important to evaluate the bone response of the ZrO_2_ surface for a similar structure and roughness as that of Ti. If we can prepare a ZrO_2_ surface with a similar structure and roughness as that of Ti, we can say something about the contribution of the surface chemical composition to the osseointegration for ZrO_2_ implant.

The molecular precursor method is a surface modifying method, i.e., an arbitrary metal oxide film coating on the ceramic and metal surfaces achieved using a metal complex liquid [18,19,20]. A thin apatite film can be deposited onto the Ti surface using the Ca–EDTA complex precursor solution [21,22]. The thickness of the thin apatite film was in the range of 0.6–0.8 μm. The molecular precursor method is a wet process [23] and can only change the chemical composition of substrates without changing the surface structure and roughness because of the very thin film coating. A thin ZrO_2_ film can be deposited on the Ti surface using the Zr-complex precursor solution. Therefore, we can achieve a similar surface structure and roughness of the ZrO_2_ surface as roughened Ti implant surface.

In this study, we tried to distinguish the effects of the surface chemical composition, Ti or ZrO_2_, from those of the surface structure and roughness between Ti and ZrO_2_ coated Ti implants in the osseointegration process. The ZrO_2_ coating fabricated by the molecular precursor method changed the surface chemical composition from Ti to ZrO_2_ without changing the surface structure and roughness of roughened SLA-Ti. The main purpose of this study is to reveal the effects of the difference in surface chemical composition on the bone response with respect to ZrO_2_ and Ti with a similar surface structure and roughness. We can make clear the difference in bone responses between Ti and ZrO_2_ surfaces. The effectiveness of the ZrO_2_ surface on the bone response will be clarified in this study. To the best of our knowledge, this is the first trial to elucidate the effectiveness of the surface chemical composition of implant materials on a bone formation without considering surface structure and roughness.

## 2. Materials and Methods

### 2.1. Specimen Preparation

Two shapes of Ti samples (JIS2 type, 99.9% mass, Furuuchi Chemical, Tokyo, Japan), disks, and rectangular plates were used in this study as shown in Figure 1. Ti disks with 12.0 mm in diameter and 1.0 mm in thickness were used for surface observation, crystal structure and atom elements analyses, phosphate-buffered saline (PBS), and simulated body fluid (SBF) immersion experiments. Ti rectangular plates with dimensions of 1.0 mm × 10.0 mm × 20.0 mm were used for zeta potential measurements and those with dimensions of 2.0 mm × 3.0 mm × 1.0 mm were for animal implantation experiments. The number of disks and rectangular plates was three for each experiment in addition to AFM measurements (*n* = 4).

A rectangular plate is better to be set into the cell of the apparatus for zeta potential measurements. Regarding the animal experiment, the bone defect is box-shaped, and a rectangular plate fits this shape of the bone defect. It is difficult to fit the disks to the bone defect. In our previous studies for surface observation or SBF immersion experiments for coating materials, we standardized the shape of the sample on a disk. So, disks were used for other experiments.

Before ZrO_2_ coating, the surfaces of the specimens were polished with a #1200 waterproof paper under running water. Then, specimens were subjected to SLA treatment. Namely, sandblasting was performed perpendicular to the titanium surface from a distance of 20 mm with 180 μm alumina particles at a 0.6 N/mm^2^ air pressure. Then, acid etching was performed on the blasted surface with a mixture of 36% hydrochloric acid (HCl) and a 96% sulfuric acid (H_2_SO_4_) solution for 3 min at 70 °C [24]. SLA-treated specimens were then ultrasonically cleaned in ethanol and distilled water for 20 min (ultrasonic cleaner; VS-100III; AS ONE, Osaka, Japan).

Ti disks and rectangular plates were coated with ZrO_2_ (ZrO_2_/Ti) using the molecular precursor method. A molecular precursor solution (25 mL) for the ZrO_2_ film (Zr complex ethanol solution; Zr ion concentration = 0.32 mmol/g; TFTECH, Tokyo, Japan) was poured on the Ti surface. It was then spin-coated using a spin coater (1H-D7; MIKASA, Tokyo, Japan) in the double step mode; first at 500 rpm for 5 s, and then at 2000 rpm for 30 s. Ti disks and rectangular specimens covered with precursor films were then heated at 550 °C in air for 30 min using a tubular furnace (EPKPO12-K; ISUZU, Niigata, Japan). A thin film coating of ZrO_2_ can be achieved on Ti disks and rectangular plates using this method.

### 2.2. Surface Characterization of the ZrO_2_ Coating

The surface structure of ZrO_2_/Ti was observed using scanning electron microscopy (SEM; SU1510; Hitachi High-Technologies, Tokyo, Japan). Au supper-coated specimens were observed at a 15 kV accelerating voltage.

The three-dimensional surface structure of ZrO_2_/Ti was observed using atomic force microscopy (AFM; Nanosurf Easyscan 2, Nanosurf AG; Liestal, Switzerland). The tapping mode was employed using a TapAL-G cantilever (Budget sensors, Bulgaria; resonance frequency = 190 kHz, spring constant = 48 N/m). The average arithmetic roughness of the three-dimensional surface (Sa) was calculated within the 1.0 μm × 1.0 μm, 2.5 μm × 2.5 μm, 5.0 μm × 5.0 μm, and 10.0 μm × 10.0 μm areas of the AFM images. The surface of roughened Ti was also observed using SEM and AFM by the same method.

X-ray diffraction was used to characterize the crystal structure of the coated ZrO_2_ thin film on Ti disks with a thin layer technique (XRD; attachment incidence angle θ = 0.3°, SmartLab; Rigaku, Tokyo, Japan). An X-ray source is Cu Kα which powers 45 kV × 200 mA. The presence of Ti or Zr atoms on Ti and ZrO_2_/Ti disks on each surface was confirmed using an electron probe micro analyzer (EPMA; JXA-8900R; JEOL, Tokyo, Japan) at an accelerating voltage of 15 kV. Cross-sectional observation was also performed. The disks were then embedded in an epoxy resin. After curing the resin, the specimens were vertically cut through the middle using a cutting machine.

The atom elements in the surface of Ti and ZrO_2_/Ti were analyzed by using X-ray photoelectron spectroscopy (XPS; AXIS-ULTRA; Kratos Analytical, Stretford, UK), with an X-ray source of Al Kα and a power of 15 kV × 10 mA.

Zeta potential is known to indicate the surface properties, especially charging behavior of the material surface. Next, we tried to determine the apparent zeta potentials using an electrokinetic analyzer (SurPASS 3; Anton Paar GmbH, Graz, Austria). For determining the zeta potential, an aqueous electrolyte solution (1.0 mM of KCl) was used at a pH level of 5.6. KCl solution passed through a thin slit channel formed by two Ti or ZrO_2_/Ti plates with identical surfaces of rectangular plate implant. The pressure-driven flow of the KCl solution gave the streaming current and potential. The zeta potential was then calculated according to the Helmholtz–Smoluchowski equation [25,26]. Furthermore, the pH-dependent curves of the zeta potential were plotted by measuring the zeta potential at any pH level adjusted by 0.05 M HCl and 0.1 M KOH. The isoelectric points of Ti and ZrO_2_/Ti were obtained from these curves. Measurements of apparent zeta potential at a pH level of 5.6 were performed three times. In these experiments, coefficient of variation was less than 3%, and it was concluded the measurements possessed high reliability. Thus, measurements of zeta potential at different pH were performed only once.

For determining the durability of ZrO_2_ coating, the ZrO_2_/Ti disks were immersed in PBS solution (pH = 7.4). The volume of PBS solution was 20 mL. The specimens were stored in 20 mL of PBS in a polypropylene bottle for 90 days. After immersion, the specimens were dried in a desiccator. The morphological changes in ZrO_2_ films were observed using SEM at an accelerating voltage of 15 kV after sputter-coating with Au.

### 2.3. SBF Immersion Experiments

As SBF, Hanks’ balanced salt solution (HBSS) was prepared without organic species and adjusted pH = 7.4 [27]. Ti and ZrO_2_/Ti disks were immersed in 20 mL of HBSS at 37 °C in a polypropylene bottle. After immersing for 6 and 24 h, the excess HBSS on the disks was removed using soft paper, and the disks were immediately dried in a desiccator. The surfaces of the Ti and ZrO_2_/Ti disks were then observed using SEM at an accelerating voltage of 15 kV after sputter-coating with Au. The crystallographic structures of the precipitates on the Ti and ZrO_2_/Ti disks were analyzed using XRD (RINT-RAPID2 MM007; Rigaku, Tokyo, Japan), with an X-ray source of Cu-Kα at a power of 40 kV × 30 mA, and FT-IR (FT/IR-430; Jasco, Tokyo, Japan) using the KBr method. The precipitated products were detached from the Ti and ZrO_2_/Ti disks for FT-IR measurements.

### 2.4. Animal Implantation Experiment

The approval of the animal implantation experiment was obtained from the Animal Experimental Ethical Guidelines of the Tsurumi University, School of Dental Medicine (certificate nos. 19A042, 20A001, 21A009, and 22A005). Overall, 12 male Wistar rats were used for animal implantation experiments. Their weight was approximately 180 g and ages are 6 weeks. The rats were housed in pairs of two per cage at 20–25 °C in a 12 h circadian light rhythm environment. They feed on water and food ad libitum during the experiment.

Ti or ZrO_2_/Ti rectangular plate implant was placed in a femur bone defect [28,29], and each rat received one implant. Overall, 3 ZrO_2_/Ti and 3 Ti rectangular plate implants were inserted for 2- and 4-week implantation periods, respectively.

For general inhalation, anesthesia using a 3% isoflurane and oxygen mixture was used. The concentration of isoflurane was reduced to 2% during surgical manipulation. The hind limb was shaved and then operating field was disinfected. Next, operation field was locally anesthetized by the injection of xylocaine. The distal surface of the hind limb was longitudinally incised, and femur was exposed. The cortical bone defect was prepared through the cortex and the medulla. A very gentle surgical technique was employed and internal cooling with a physiological saline solution was continuously introduced during the defect preparation. The size of the defect was 1 mm × 2 mm. A rectangular plate implant was press fitted into the bone defect as shown in Figure 2. The muscle tissue and skin were separately sutured using non-absorbable. After surgery, antibiotics (benzyl penicillin G procaine, 3,000,000 U/kg) were subcutaneously injected.

The rats were euthanized at 2 and 4 weeks after implantation using carbon dioxide gas to harvest the rectangular plate implants and the surrounding femoral bone.

After removing the excess tissue, the specimens were fixed in a 10% neutral buffered formalin solution (pH 7.4). Then, tissue blocks including implants were dehydrated through a gradient series of ethanol. Subsequently, they were embedded in methyl methacrylate. A cutting–grinding technique was applied to make non-decalcified thin sections. Cutting of the tissue blocks was performed in a direction perpendicular to the axis of the implants. After cutting and grinding by EXAKT-Cutting Grinding System (BS-300CP band system and 400CS microgrinding system; EXAKT, Norderstedt, Germany), thin sections in which thickness was approximately 50–70 μm were obtained.

Thin sections were double stained with methylene blue and basic fuchsin. Histological and histomorphometrical evaluation was performed using a light microscope (Eclipse N*i*, Nikon, Tokyo, Japan, magnifications of ×40 and ×100). Bone response towards rectangular plate implant was histologically evaluated. As a histomorphometric analysis, the bone-to-implant contact (BIC) ratio and the bone mass (BM) around the implant were measured within the regions of interest (ROI). The ROIs were determined in accordance with a previous study [30], as illustrated in Figure 3. The BIC ratio was defined as the percentage of the length of direct bone–implant contact along with the implant surface. BM was calculated as the percentage of newly formed bone around the implant. The BIC ratio and BM values were obtained by the image analysis (WinROOF, Visual System Division Mitani, Tokyo, Japan).

### 2.5. Statistical Analysis

All data were calculated with the help of SPSS for Windows (SPSS Statics 17.0; SPSS, Chicago, IL, USA). The results of the surface roughness from AFM measurements were compared using the student *t*-test. A *p*-value < 0.05 was considered to be significant. The results of the BIC ratio and BM values were statistically analyzed by one-way analysis of variance (ANOVA) and Tukey’s test for multiple comparisons (*p* < 0.05).

## 3. Results

### 3.1. Surface Characterization

Figure 4 shows the SEM profiles of the Ti and ZrO_2_/Ti disk surfaces. No difference was observed in the surface shapes between the Ti and ZrO_2_/Ti disks. Both surfaces showed micro-scale dimples (arrows) on the roughened surface, which were formed by irregular roughness. Additionally, the detachment and defect of the ZrO_2_ coating film were not observed.

Figure 5 shows the AFM profiles of the Ti and ZrO_2_/Ti disk surfaces. The obtained three-dimensional structure was almost identical on both surfaces. Moreover, there was no significant difference in the arithmetic average roughness of the three-dimensional surface (*p* > 0.05) (Table 1).

Figure 6 shows the EPMA mapping of ZrO_2_/Ti disks. The presence of ZrO_2_ coating was confirmed by the element mapping of Zr. Cross-sectional observation also identified the presence of Zr in the coating film (arrows in Figure 6b).

Figure 7 shows the XPS results. The O1s peak was observed at 526.5 eV owing to the presence of TiO_2_ and ZrO_2_. The peaks attributed to Zr3d_2/3_ and Zr3d_5/2_ were observed at 179 and 181 eV, respectively. The peaks of Ti2p_3/2_ and Ti2p_1/2_ of the Ti substrate were observed at 461 and 455 eV, respectively, because of the low thickness of the ZrO_2_ layer. Thus, the presence of the ZrO_2_ coating on the surface of Ti disks was confirmed.

Figure 8 shows the XRD patterns of the ZrO_2_ films on the Ti disk. The deposited coating with ZrO_2_ structures was confirmed. Corresponding to monoclinic zirconia, a peak was observed at 27.5°, whereas peaks at 36.1°, 37.7°, 41.2°, 54.3°, 62.6°, 69.0°, and 76.7° were observed for tetragonal zirconia. Additionally, Ti peaks were also observed.

The apparent zeta potential of ZrO_2_/Ti was higher than that of Ti at a pH level of 5.6 (Table 2). Figure 9 shows the pH-dependent curves of the zeta potential at different pH levels. The isoelectric points of Ti and ZrO_2_/Ti obtained from Figure 8 are also listed in Table 2. The isoelectric point of ZrO_2_/Ti was higher than that of Ti.

Figure 10 shows the SEM pictures of the surfaces of ZrO_2_/Ti disks after PBS immersion. No differences were observed in the surface morphologies of ZrO_2_/Ti disks. The formation of cracks and pinholes was not observed on the surface.

### 3.2. SBF Immersion Experiments

Figure 11 shows the SEM images of the surfaces of each specimen after 6 and 24 h HBSS immersion. Crystals were deposited on the surfaces of all specimens. However, the growth of crystals was more on the surfaces of ZrO_2_/Ti disks compared to that on the surfaces of Ti disks after 6 and 24 h of immersion.

Figure 12 shows the XRD and FT-IR profiles of the precipitated crystals 24 h after immersion. The peaks of the XRD patterns derived from apatite structures were identified at approximately 23.0°, 26.0°, 28.5°, 32.0°, 33.5°, and 46.5°. The peaks of the FT-IR spectra derived from phosphate groups were detected in the ranges of 550–600 and 900–1200 cm^−1^, whereas those from carbonyl groups were detected at approximately 1200 cm^−1^. Therefore, the precipitated crystals were identified as carbonate-containing hydroxyapatite.

### 3.3. Histological and Histomorphometric Evaluations

The rats remained in good health during the experiment. No clinical signs of inflammation or adverse tissue reactions were observed when the animals were euthanized, and all implants were still in situ.

Figure 13 and Figure 14 show the histological appearances of the ZrO_2_/Ti and Ti implants. No inflammatory cells’ permeation was observed. New bone was formed around the rectangular plate implants at 2 and 4 weeks after implantation. The bone remodeling process and mature bone formation were observed. Bone-to-implant bonding was detected for the ZrO_2_/Ti and Ti implants. However, some gaps were recognized between the bone and the ZrO_2_/Ti and Ti implants. Overall, Ti implants showed more gaps between the bone and the implant surface.

Table 3 lists the results of the BIC ratio and BM values after implantation. The BIC ratio and BM values for ZrO_2_/Ti implants at 2 weeks were significantly higher than those for Ti implants (*p* < 0.05). At 4 weeks of implantation, there were no significant differences between the Ti and ZrO_2_/Ti implants in terms of the BIC ratio and BM values (*p* > 0.05). The BIC ratio and BM values for the Ti implants at 4 weeks were significantly higher than those for Ti implants at 2 weeks (*p* < 0.05). However, no significant differences were observed between the BIC ratio and BM values of ZrO_2_/Ti implants after 2 and 4 weeks (*p* > 0.05).

## 4. Discussion

In this study, we prepared a ZrO_2_-coated Ti surface with a similar structure and roughness as that of a Ti surface using the molecular precursor method and evaluated the bone response toward ZrO_2_ thin films. It was revealed that the ZrO_2_ surface with the same structure and roughness as those of Ti promotes osteogenesis equivalent to or better than that of Ti in the early stages of bone formation.

The surface structure and roughness analysis of ZrO_2_/Ti using SEM and AFM confirmed that the surface appearances and roughness of Ti and ZrO_2_/Ti were almost similar. The molecular precursor method can change the surface chemical composition without changing the surface structure and roughness. EPMA showed that ZrO_2_ films were uniformly observed on all parts of the substrate.

Some studies have been reported on the osseointegration of ZrO_2_ dental implants in comparison with that of Ti implants [31,32]. Gahlert et al. [33]. investigated the BIC ratio and peri-implant bone density of ZrO_2_ implants with a rough acid-etched surface structure and roughness and compared it with Ti-SLA implants in the maxilla of pigs. They reported that there was no difference in the bone responses of ZrO_2_ implants and Ti-SLA controls. However, an unacceptable difference was observed in the surface structure and roughness of the ZrO_2_ and Ti implants using confocal three-dimensional white light microscopy. The Sa values with roughened ZrO_2_ implants were higher than those of Ti-SLA implants. Thus, a majority of the studies on bone responses have not focused on the same surface structure and roughness between the ZrO_2_ and Ti specimens. This present study is the first study comparing the bone responses of the Ti and ZrO_2_ implants with almost similar surface structure and roughness.

The XRD patterns of the ZrO_2_ films on the Ti disk revealed that the deposited coating comprised monoclinic and tetragonal zirconia. The broad peaks in XRD and FR-IR are due to the low crystallinity of precipitated apatite. It is well-known that the crystal structure of ZrO_2_ changes with temperature. Therefore, a stabilized agent, such as yttria, is often used to stabilize tetragonal zirconia at room temperature. However, the molecular precursor solution did not contain the stabilized agents. The presence of tetragonal zirconia in this study was due to the extreme thinness of the ZrO_2_ films. The atoms in ZrO_2_ films can easily move at low energies, which might reduce the phase transition temperature. This provided different results from the bulk of ZrO_2_.

Kokubo et al. [34] reviewed the history of SBF, the correlation of the ability of apatite to form on various materials in SBF with their in vivo bone bioactivities, and some examples of the development of novel bioactive materials based on apatite formation in SBF and concluded that examination of apatite formation on material in SBF is useful for predicting the in vivo bone bioactivity of a material. Thus, we first employed the SBF immersion experiments and then performed the animal experiments. As a result, SBF immersion experiments suggested that the surface property of ZrO_2_ promotes osteogenesis equivalent to or better than that of Ti in the early stages of bone formation. This result corresponded with that observed in the animal experiments.

The animal study suggested that with the same surface structure and roughness as that of Ti rectangular plate implants, the ZrO_2_ surface exhibited the promotion of new bone formation surrounding the implants. In this study, the apparent zeta potential measurement showed that the value of ZrO_2_/Ti was higher than that of Ti at any pH level. It is inferred that the electrostatic repulsion force of the ZrO_2_ against bone proteins, such as osteocalcin and osteopontin, which are negatively charged, may be smaller than that of Ti. Thus, these reactions may promote bone formation on ZrO_2_.

In the animal experiments, rectangular plate implants were inserted into the femurs of rats, and 2- and 4-week implantation periods were evaluated because we intended to observe the early stage of bone formation. A significantly higher value in BIC and BM for ZrO_2_/Ti was observed. Albrektsson et al. [35] argued that osseointegration corresponded to approximately 60% bone contact for Ti implants. In this study, the BIC ratio of the Ti implant was 50–70%, whereas that of the ZrO_2_/Ti implant was approximately 75%. In other words, the BIC ratio of the ZrO_2_/Ti implant was expected to be above the limit prescribed by Albrektsson et al.

This is a new approach to distinguishing the effects of surface chemistry from those of surface structure and roughness between ZrO_2_ and Ti in the osseointegration process. This study is basic research to evaluate the bone response of the ZrO_2_ surface with the same structure and roughness as that of Ti for elucidating the osseointegration of ZrO_2_ implants. In contrast, it was suggested that ZrO_2_-coated Ti implants also have a sufficient clinical value from the viewpoint of BIC ratio and BM value. Non-metallic implants have been reported to exhibit the risks of fracture [36]. Therefore, ZrO_2_ is often used as implant abutments only clinically. However, ZrO_2_-coated Ti implants can resolve this fracture problem because of the better mechanical properties of Ti. Moreover, it is expected that a ZrO_2_ film preventing corrosion on the Ti substrate will lead to a decrease in allergies and hypersensitivity problems.

Two-step rotations for spin coating were determined according to a previous report by Sato et al. [18]. The first step of spin coating, 500 rpm (low speed), was the setting for spreading the precursor solution over the entire substrate. The second step, 2000 rpm (high speed), was aimed to blow away the excess coating precursor solution and achieve a film thickness of fewer than 1 μm.

In these experiments, we inserted the rectangular plates into the cortical limb bone of rats. This is the basic model for evaluating the bone response to surface-modified materials as previously reported [28,29,30]. In dental clinics, cylindrical or screw-type implants were normally used. We already performed the thin film coating on tooth-shaped or cylindrical implants in the molecular precursor method using the spin-coating technique [37,38]. Moreover, bone responses after the implantation into the jaw bone should be examined. We also developed a mandibular bone defect model in rats [15]. In the next series of our experiments, ZrO_2_ coating was deposited on the cylindrical or screw-type implants and assessed the bone formation after the implantation of ZrO_2_ coating implants into the bone defects in the maxilla or mandible.

According to the results of this study, it is suggested that the ZrO_2_ surface with the same surface structure and roughness as that of Ti implants can promote bone formation. In other words, ZrO_2_ implant surfaces are required to have the same roughness as that Ti implants despite the difficulty in processing. Moreover, sandblasting Y-TZP surfaces caused a problem of phase transformation [39]. Hirota et al. prepared a unique roughened surface of Y-TZP by nano-pulsed laser irradiation without phase transformation, thereby enhancing the bone response [28]. Thus, it is important to have efficient roughened ZrO_2_ surfaces, such as those in Ti implants, without phase transformation to achieve ZrO_2_ osseointegration.

## 5. Conclusions

In this study, we evaluated the bone response of the ZrO_2_ surface with the same structure and roughness as those observed with Ti rectangular plate implants. The molecular precursor method, which can deposit any shape without changing the material surface structure and roughness, produced a thin ZrO_2_ film on the rough Ti surface. It was revealed that ZrO_2_ with a similar surface structure and roughness as that of roughened Ti rectangular plate implants promoted osteogenesis equivalent to or better than that of Ti in the early bone formation stage. We can make clear the effectiveness of the surface chemical composition of the ZrO_2_ surface on a bone formation without considering the factors of surface structure and roughness. The osseointegration mechanism will be clarified by investigating the ZrO_2_ surface chemical composition details.

## Figures and Tables

**Figure 1 nanomaterials-12-02478-f001:**
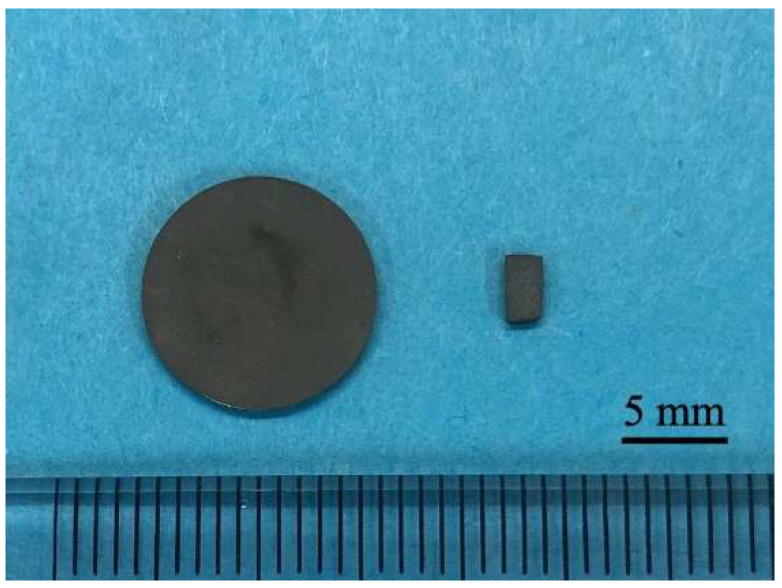
The picture of titanium disk and rectangular plate.

**Figure 2 nanomaterials-12-02478-f002:**
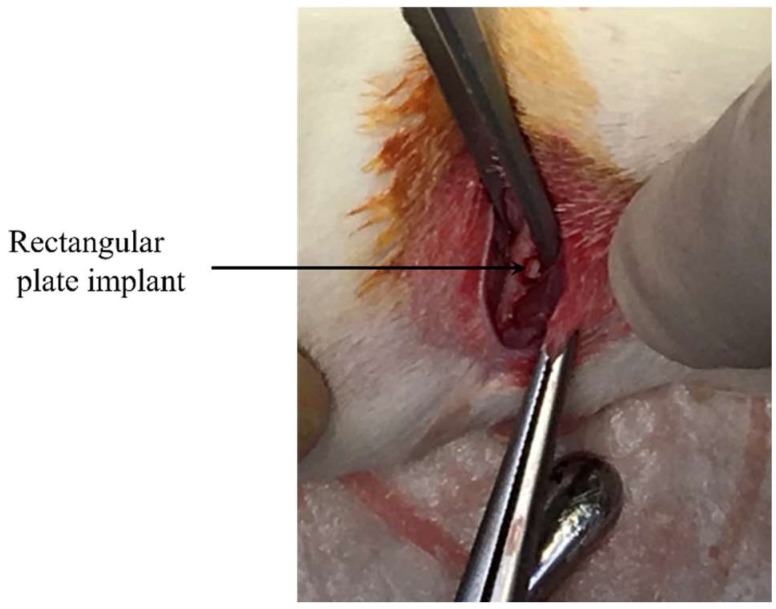
Implantation of rectangular plate implant into femur bone defect.

**Figure 3 nanomaterials-12-02478-f003:**
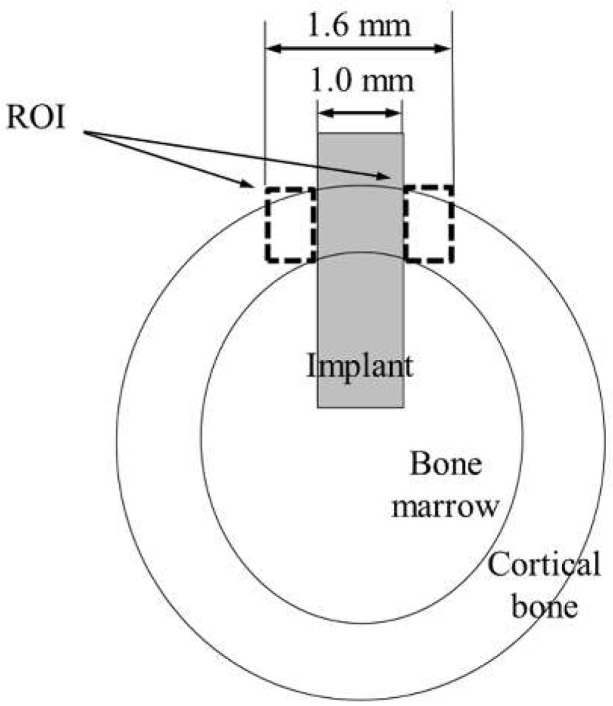
Schematic of the ROI.

**Figure 4 nanomaterials-12-02478-f004:**
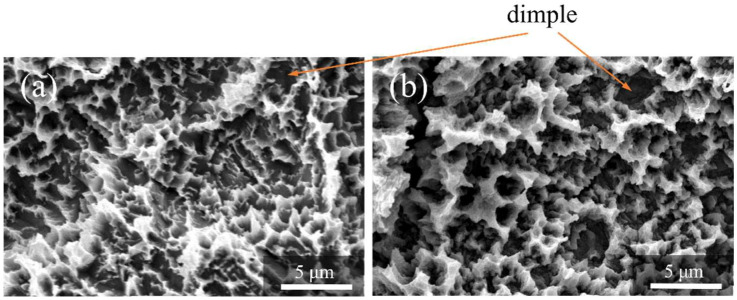
SEM profiles of the surfaces of the (**a**) Ti and (**b**) ZrO_2_/Ti disks.

**Figure 5 nanomaterials-12-02478-f005:**
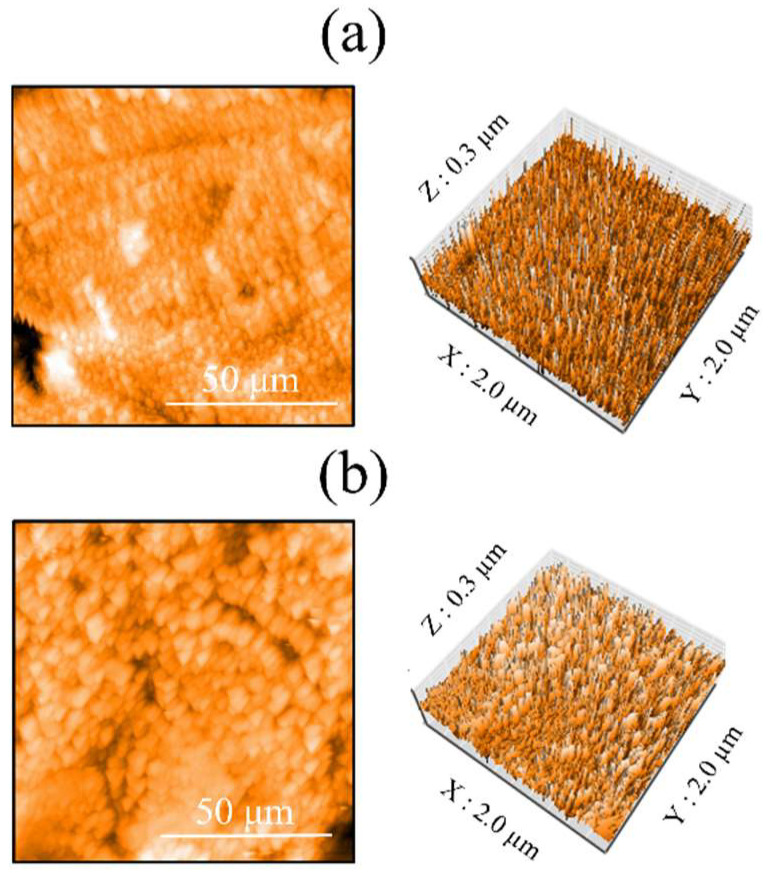
AFM profiles of the surfaces of the (**a**) Ti and (**b**) ZrO_2_/Ti disks.

**Figure 6 nanomaterials-12-02478-f006:**
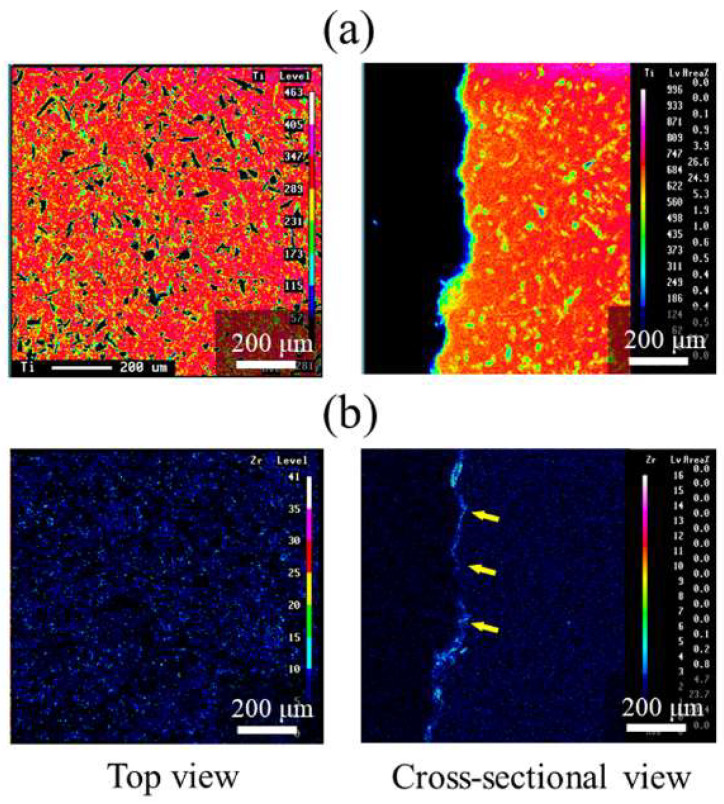
EPMA mappings of ZrO_2_/Ti disks. (**a**) Ti and (**b**) Zr mappings. The arrows on (**b**) indicate Zr elements.

**Figure 7 nanomaterials-12-02478-f007:**
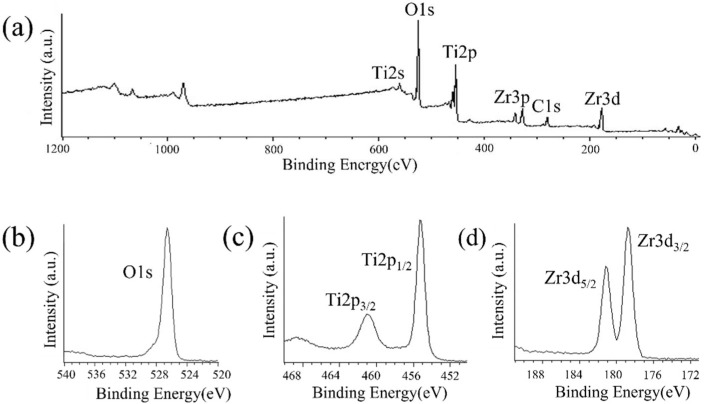
XPS profiles of ZrO_2_/Ti disk surfaces. (**a**) XPS broad spectrum of the ZrO_2_/Ti surface. (**b**) O1s, (**c**) Ti2p, and (**d**) Zr3d spectra of ZrO_2_/Ti surfaces.

**Figure 8 nanomaterials-12-02478-f008:**
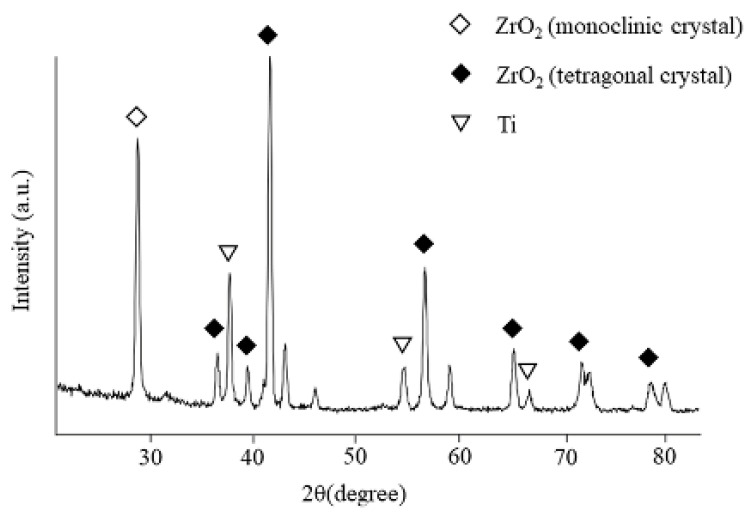
XRD patterns of the films on ZrO_2_/Ti disks.

**Figure 9 nanomaterials-12-02478-f009:**
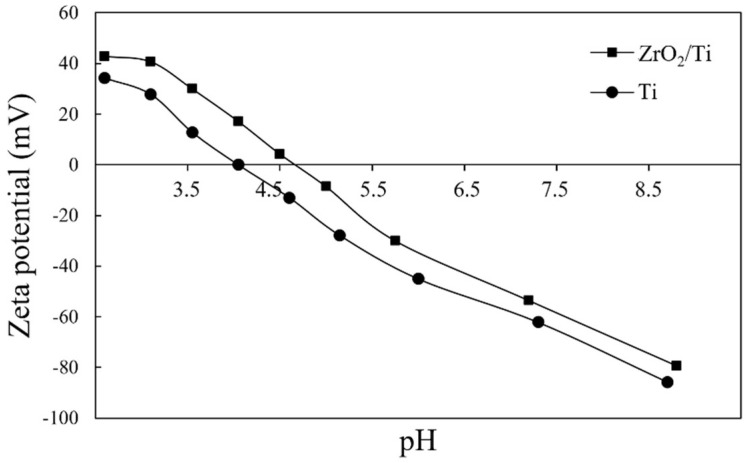
pH-dependent curves of the apparent zeta potentials of Ti and ZrO_2_/Ti at any pH level.

**Figure 10 nanomaterials-12-02478-f010:**
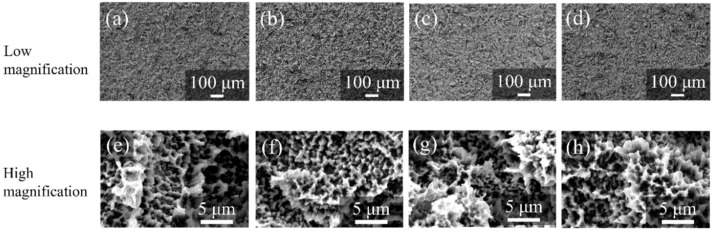
SEM profiles of the surfaces of ZrO_2_/Ti disks after PBS immersion. (**a**,**e**) 7, (**b**,**f**) 14, (**c**,**g**) 30, and (**d**,**h**) 90 days after immersion. (Magnification of ×100 and ×5000).

**Figure 11 nanomaterials-12-02478-f011:**
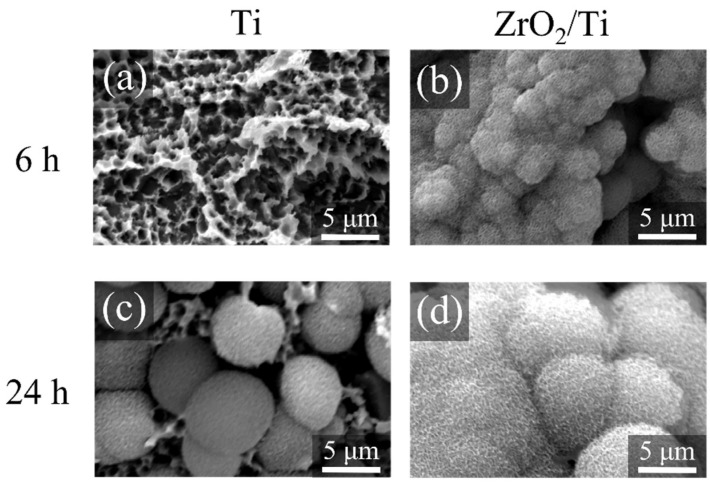
SEM profiles of the surfaces of the Ti and ZrO_2_/Ti disks after SBF immersion. (**a**,**c**) Ti and (**b**,**d**) ZrO_2_/Ti disks. (**a**,**b**) 6 and (**c**,**d**) 24 h after immersion.

**Figure 12 nanomaterials-12-02478-f012:**
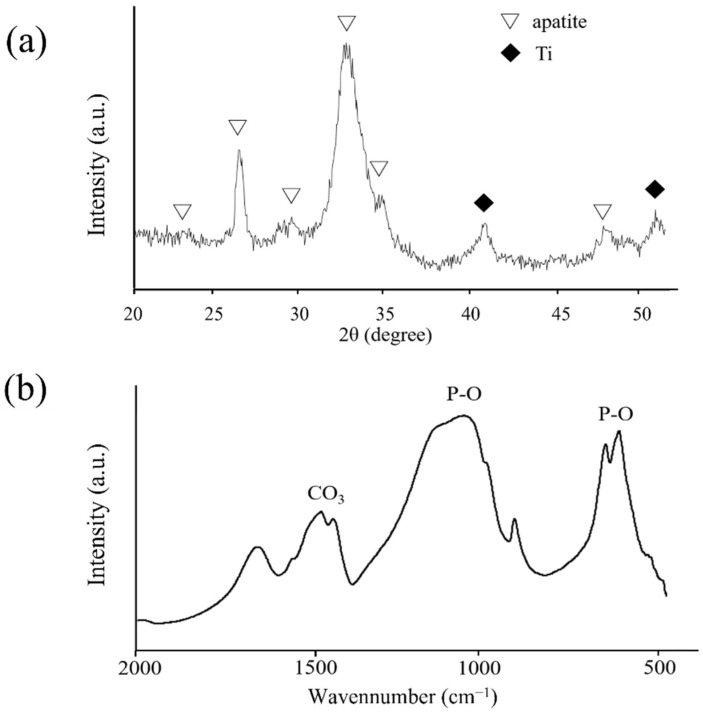
(**a**) XRD profiles of the deposited crystals on ZrO_2_/Ti disks after 14 days of HBSS immersion. (**b**) FT-IR spectrum of the deposited crystals on ZrO_2_/Ti disks after 14 days of HBSS immersion.

**Figure 13 nanomaterials-12-02478-f013:**
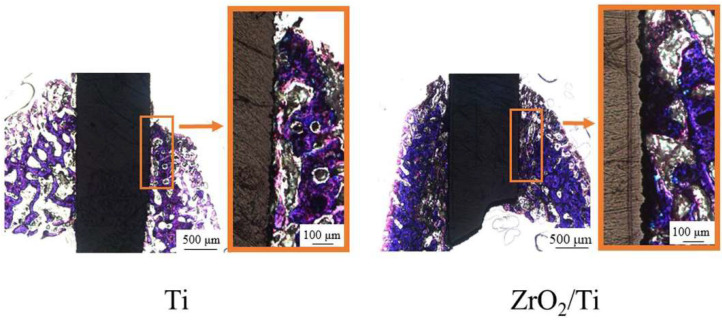
Histological appearances after 2 weeks of implantation into the femur bone defects of rats (magnification of ×40 and ×100).

**Figure 14 nanomaterials-12-02478-f014:**
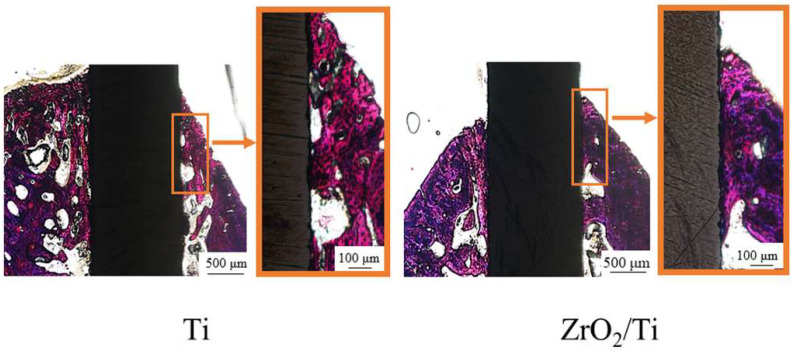
Histological appearances after 4 weeks of implantation into the femur bone defects of rats (magnification of ×40 and ×100).

**Table 1 nanomaterials-12-02478-t001:** Three-dimensional surface roughness (Sa) of Ti and ZrO_2_/Ti disks.

		Square Areas			
		10 μm × 10 μm	25 μm × 25 μm	50 μm × 50 μm	100 μm × 100 μm
Sa (µm)	Ti	0.38 (0.12) ^a^	0.82 (0.22) ^b^	1.11 (0.12) ^c^	1.30 (0.13) ^d^
	ZrO_2_/Ti	0.54 (0.19) ^a^	0.67 (0.14) ^b^	1.05 (0.41) ^c^	1.32 (0.03) ^d^

Values in brackets are SD. Same superscripts indicate no significant differences in each square areas between the Ti and ZrO_2_/Ti groups.

**Table 2 nanomaterials-12-02478-t002:** Apparent zeta potentials at a pH level of 5.6, and the isoelectric points of the Ti and ZrO_2_/Ti specimens.

Specimen	Zeta Potential (mV)	Isoelectric Point
Ti	−33.25 (0.08)	4.05
ZrO_2_/Ti	−20.69 (0.55)	4.70

Values in brackets are SD.

**Table 3 nanomaterials-12-02478-t003:** Percentage of BIC ratio and BM values.

Implantation Period	Specimen	BIC Ratio (%)	BM (%)
2 weeks	Ti	58.8 (1.33) ^a, A^	64.0 (0.69) ^a, A^
	ZrO_2_/Ti	72.9 (0.70) ^b, C^	73.5 (2.31) ^b, C^
4 weeks	Ti	72.5 (2.58) ^c, B^	71.5 (2.13) ^c, B^
	ZrO_2_/Ti	73.6 (3.08) ^c, C^	72.1 (2.82) ^c, C^

Values in brackets are SD. The same superscripts indicate no significant differences. Different small letters indicate significant differences between Ti and ZrO_2_/Ti in the same implantation period. Different capital letters indicate significant differences between the values obtained at 2 and 4 weeks in the same specimen.

## Data Availability

Data presented in this article are available on request from the corresponding author.

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
