# Peer review of "Influence of the Surface Chemical Composition Differences between Zirconia and Titanium with the Similar Surface Structure and Roughness on Bone Formation"

_nanomaterials, 2022, doi:10.3390/nano12142478_

Round 1

Reviewer 1 Report

The manuscript reports the effects of the surface chemistry from those of the surface topography between ZrO2 and Ti in the osseointegration process. The ZrO2 coating fabricated by the molecular precursor method changed the chemical composition without changing the surface topography of SLA-Ti. The results demonstrated that ZrO2 with the same surface topography as that of roughened Ti implants promoted osteogenesis equivalent to or better than that of Ti in the early stages of bone formation. These contents are interesting and fall within the scope of NANOMATERIALS. I recommend its acceptance for publication after some revisions.

1) The ABSTRACT should have key quantitative results for the readers.

2) Figure 7 should have error bars and the repeated times should be indicated.

3) Figure 8 and Figure 9, please provide images under a BIGGER magnifications, for nanomaterials, the scale bar of 100 μm is too large, which at least 1 to 5 μm for more information at nanoscale.

4) The potential influence of working parameters using the spin coater on the resultant products should be discussed.  

Author Response

List of the point by point responses to the reviewer’s comments

Reviewer 1

Comments and Suggestions for Authors: The manuscript reports the effects of the surface chemistry from those of the surface topography between ZrO2 and Ti in the osseointegration process. The ZrO2 coating fabricated by the molecular precursor method changed the chemical composition without changing the surface topography of SLA-Ti. The results demonstrated that ZrO2 with the same surface topography as that of roughened Ti implants promoted osteogenesis equivalent to or better than that of Ti in the early stages of bone formation. These contents are interesting and fall within the scope of NANOMATERIALS. I recommend its acceptance for publication after some revisions.

1) The ABSTRACT should have key quantitative results for the readers.

Answer: According to the comment, we added the quantitative results in abstract section as following (lines 18-22).

Ti and ZrO2-coated Ti implant was placed into the femur bone defect. After 2- and 4 weeks implantation, histomorphometrical observation revealed that the bone-to-implant contact ratio and the bone mass values for ZrO2-coated Ti implants inserted into the femur bone defects of the rats at 2 weeks were significantly higher than those for Ti implants (P < 0.05).

2) Figure 7 (Figure 9: new figure number) should have error bars and the repeated times should be indicated.

Answer: First we measured the apparent zeta potential at a pH level of 5.6 (Table 2). In these measurements, we checked the validation of the measurements. We performed three times measurements for each sample. The coefficient of variation was less than 3 %, and it was concluded the measurements possessed high reliability. Thus, measurements of zeta potential at different PH was done only once. This is the normal method for obtain the zeta potential at different pH.

                 We added the number of measurements for zeta potential in Table 2 and added some comments as following at lines 151-154 in Materials and Methods section. We added the acknowledgment for the measurements of zeta potential.

Measurements of apparent zeta potential at a pH level of 5.6 were performed three times. In this experiments, coefficient of variation was less than 3 %, and it was concluded the measurements possessed high reliability. Thus, measurements of zeta potential at different pH was done only once.

Acknowledgment

The authors are grateful to Mr. Ryukichi Takagi and Mr. Yuki Nakano in Anton Paar Japan K.K. for measurements of zeta potential.

3) Figure 8 and Figure 9, please provide images under a BIGGER magnifications, for nanomaterials, the scale bar of 100 μm is too large, which at least 1 to 5 μm for more information at nanoscale.

Answer: We added and replaced higher magnification SEM images for ZrO2/Ti and Ti disks after PBS and SBF immersion as show in Figure 10 and 11 (new figure number), recpectively.

Figure 10. SEM profiles of the surfaces of ZrO2/Ti disks after PBS immersion. (a) (e) 7, (b) (f) 14, (c) (g) 30, and (d) (h) 90 days after immersion. (magnification of ×100 and ×5000)

Figure 11. SEM profiles of the surfaces of the Ti and ZrO2/Ti disks after SBF immersion. (a), (c) Ti and (b), (d) ZrO2/Ti disks. (a), (b) 6 and (b), (d) 24 h after immersion.

4) The potential influence of working parameters using the spin coater on the resultant products should be discussed. 

Answer: According to the comment, we added new sentences in the Discussion section at lines 392-396.

Two step conditions for spin coating was determined according to previous report by Sato et al [18]. The first step of spin coating, 500 rpm (low-speed), was the set for spreading the precursor solution over the entire substrate. The second step, 2000 rpm (high-speed), was aimed to blow away the excess coating precursor solution and to achieve a film thickness of less than 1 μm.

[18] Sato, M.; Hara, H.; Nishide, T.; Sawada, Y. A water-resistant precursor in a wet process for TiO2 thin film formation. J. Mater. Chem. 1996, 6, 1767-1770.

Reviewer 2 Report

Dear authors,

According to my peer review, the following manuscript entitled - Influence of the surface chemistry differences between zirconia and titanium with the same surface topographies on bone formation - addresses a pertinent topic and fall within the scope of Nanomaterials Journal. Although recognising some merits of this research work several aspects/comments were highlighted in this peer review for a future clarification/review by authors:

1)    An updated review of the topic, surface osseointegration of zirconia implants, was properly provided in the Introduction. Congratulations.

2)    Authors should better clarify roughness and topography. Both concepts are similar but not equal. In your manuscript both seem indistinct (from line 42 to 55)

3)    Sample size was not mentioned by authors. How many disks and rectangular plates were used? Please explain and discuss why these two sample geometries were chosen?

4)    Authors are invited to attach images of the disk and rectangular plates used in your research work.

5)    Is the molecular precursor method applicable to an implant instead of a disk? 

6)    From 2.2 (in vitro biocompatibility evaluation by SBF immersion) to 2.3 (in vivo study) authors no more refer disks or plates. Instead, authors mention “implants”. What kind of implants were used? This might be confusing for readers. Which type of sample was inserted in bone defects? Authors should clarify this point. 

7)    Once again, authors are invited to attach images of the in vivo study. What was the criteria for choosing this animal model? In my opinion is a poor choice considering the aim of this research and expressed in the title: “Influence of the surface chemistry differences between zirconia and titanium with the same surface topographies on bone formation” This aspect justifies my position about the scientific soundness of your work.

8)    Why only 2- and 4-week period of osseointegration was considered instead of the common 10 to 12 weeks?

9)    In discussion authors mention that “In this study, we prepared ZrO2-coated Ti implants with the same roughened surface topography as that of Ti implants using the molecular precursor method and evaluated the bone response toward ZrO2 thin films.” This sentence lacks precision. Authors prepared implants?disks?rectangular plates? with similar roughness as presented in Table 1

10)  Limitations of the research work should be better discussed in Discussion.

According to the previous points a major review is proposed for a future re-submission.

Author Response

List of the point by point responses to the reviewer’s comments

Reviewer 2 Comments and Suggestions for Authors

                 According to my peer review, the following manuscript entitled - Influence of the surface chemistry differences between zirconia and titanium with the same surface topographies on bone formation - addresses a pertinent topic and fall within the scope of Nanomaterials Journal. Although recognising some merits of this research work several aspects/comments were highlighted in this peer review for a future clarification/review by authors:

1)    An updated review of the topic, surface osseointegration of zirconia implants, was properly provided in the Introduction. Congratulations.

Answer: Thank you for your comments.

2)    Authors should better clarify roughness and topography. Both concepts are similar but not equal. In your manuscript both seem indistinct (from line 42 to 55)

Answer:

We used “surface structure and roughness” instead of “surface topography” in the whole text including title, and added next sentences at line 46-47.

Surface roughness is popularly used parameter for evaluating the material’s surface.

3)    Sample size was not mentioned by authors. How many disks and rectangular plates were used? Please explain and discuss why these two sample geometries were chosen?

Answer: We described as following in Materials and Methods section, but it was not clear for the readers.

“Disk specimens were used for the characterization of the ZrO2 coating and the in vitro immersion experiments in phosphate-buffered saline (PBS) and simulated body fluid (SBF). The rectangular plate specimens were used for animal experiments.”

We placed next sentences in Materials and Methods section at lines 84-98.

Two shapes of Ti samples (JIS2 type, 99.9% mass, Furuuchi Chemical Corp., Tokyo, Japan), disks and rectangular plates were used in this study as shown in Figure 1. Ti disks (12.0 mm in diameter and 1.0 mm in thickness) were used for surface observation, crystal structure and atom elements analyses, phosphate-buffered saline (PBS) and simulated body fluid (SBF) immersion experiments. Ti rectangular plates with dimension of 1.0 mm × 10.0 mm × 20.0 mm were used for zeta potential measurements and those with dimension of 2.0 mm × 3.0 mm × 1.0 mm were for animal implantation experiments. The number of disks and rectangular plates were three for each experiment besides AFM measurements. (n=4)

                 Rectangular plate is better to be set into the cell of apparatus for zeta potential measurements. Regarding the animal experiment, bone defect is box shape and rectangular plate fits this shape of bone defect. It is difficult to fit the disks to bone defect. In our previous studies for surface observation or SBF immersion experiments for coating materials, we standardized the shape of the sample on a disk. So disks were used for other experiments.

4)    Authors are invited to attach images of the disk and rectangular plates used in your research work.

Answer. We attached the picture of titanium disk and rectangular plate as Figure 1.

Figure 1. The picture of titanium disk and rectangular plate.

5)    Is the molecular precursor method applicable to an implant instead of a disk? 

Answer. We already applied molecular precursor method to cylindrical and tooth shaped implants. The molecular precursor method can form uniform coating films on substrates of any shapes and sizes. We added the next sentences at lines 400-401 and new references.

We already preformed the thin film coating to tooth-shaped or cylindrical implants in molecular precursor method using spin-coating technique [37-38].

New References

[37] Kano, T.; Yamamoto, R.; Miyashita, F.; Komatsu, K.; Hayakawa, T.; Sato, M.; Oida, S. Regeneration of periodontal ligament for apatite-coated tooth-shaped titanium implants with and without occlusion using rat molar model. J. Hard. Tissue. Biology. 2012, 21, 189-202.

[38]  Hirota, M.; Hayakawa, T.; Ohkubo, C.; Sato, M.; Hara, H.; Toyama, T.; Tanaka, Y. Bone responses to zirconia implants with a thin carbonate-containing hydroxyapatite coating using a molecular precursor method. J. Biomed. Mater. Res. B. Appl. Biomater. 2014, 102B, 1277-1288.

6)    From 2.2 (in vitro biocompatibility evaluation by SBF immersion) to 2.3 (in vivo study) authors no more refer disks or plates. Instead, authors mention “implants”. What kind of implants were used? This might be confusing for readers. Which type of sample was inserted in bone defects? Authors should clarify this point. 

Answer: As answered to above comments, we used disks for surface observation, crystal structure and atom elements analyses, PBS and SBF immersion experiments, and rectangular plates for zeta potential measurements and animal experiments. We added the next sentences at lines 84-92, and we changed “implant” to “rectangular plate implant” in animal experiment section.

Two shapes of Ti samples (JIS2 type, 99.9% mass, Furuuchi Chemical Corp., Tokyo, Japan), disks and rectangular plates were used in this study as shown in Figure 1. Ti disks (12.0 mm in diameter and 1.0 mm in thickness) were used for surface observation, crystal structure and atom elements analyses, phosphate-buffered saline (PBS) and simulated body fluid (SBF) immersion experiments. Ti rectangular plates with dimension of 1.0 mm × 10.0 mm × 20.0 mm were used for zeta potential measurements and those with dimension of 2.0 mm × 3.0 mm × 1.0 mm were for animal implantation experiments. The number of disks and rectangular plates were three for each experiment besides AFM measurements. (n=4)

7)    Once again, authors are invited to attach images of the in vivo study. What was the criteria for choosing this animal model? In my opinion is a poor choice considering the aim of this research and expressed in the title: “Influence of the surface chemistry differences between zirconia and titanium with the same surface topographies on bone formation” This aspect justifies my position about the scientific soundness of your work.

Answer: We attached the images of the in vivo study as Figure 2. In our previous studies for the evaluation of bone response to surface-modified rectangular plate implants, we used femur or tibia bone defect model as a basic research of the material surface. [28, 30] We cited the new reference regarding this model [29]. As described to the answer to your comment No.10), we have a plan for evaluating bone response towards cylindrical or screw-type implants after the implantation into the bone defects in maxilla or mandible.

Figure 2. Implantation of rectangular plate implant into femur bone defect.

New References

[29] Suzuki,T.; Hayakawa, T.; Kawamoto, T.; Gomi, K. Bone response of TGF-β2 immobilized titanium in a rat model. Dent. Mater. J. 2014, 33, 233-241.

8)    Why only 2- and 4-week period of osseointegration was considered instead of the common 10 to 12 weeks?

Answer: We want to observe the early stage of bone response after the implantation. We added the next sentences at lines 373-376 in Discussion section.

In the animal experiments, rectangular plate implants were inserted into the femurs of rats, and 2- and 4-weeks implantation periods were evaluated because we intended to observe the early stage of bone formation. A significant higher values in BIC and BM for ZrO2/Ti was observed.

9)    In discussion authors mention that “In this study, we prepared ZrO2-coated Ti implants with the same roughened surface topography as that of Ti implants using the molecular precursor method and evaluated the bone response toward ZrO2 thin films.” This sentence lacks precision. Authors prepared implants? disks? rectangular plates? with similar roughness as presented in Table 1

Answer: We changed this sentence as following at lines 326-330 according to your comment. And, we changed “surface topography” to “surface structure and roughness”.

In this study, we prepared ZrO2-coated Ti surface with the similar structure and roughness as that of Ti surface using the molecular precursor method and evaluated the bone response toward ZrO2 thin films. It was revealed that the ZrO2 surface with the same structure and roughness as those of Ti promotes osteogenesis equivalent to or better than that of Ti in the early stages of bone formation.

10)  Limitations of the research work should be better discussed in Discussion.

Answer: We added the next sentences in Discussion section at lines 397-406. 

In these experiments, we inserted the rectangular plates into the cortical limb bone of rats. This is the basic model for evaluating the bone response to surface-modified materials as previously reported [28-30]. In dental clinics, cylindrical or screw-type implants were normally used. We already preformed the thin film coating to tooth-shaped or cylindrical implants in molecular precursor method using spin-coating technique [37-38]. Moreover, bone responses after the implantation into the jaw bone should be examined. We also developed mandibular bone defect model in rat [15]. As a next series of our experiments, ZrO2 coating was deposited on the cylindrical or screw-type implants and assed the bone formation after the implantation of ZrO2 coating implants into the bone defects in maxilla or mandible.

.

Reviewer 3 Report

Yoshiki et al. reported ZrO2-coated Ti implants with the same topography as that of roughened Ti substrates using the molecular precursor method and study the Influence of the surface chemistry differences for osseointegration. The presentation and organization of the manuscript are unsatisfactory and the conclusions are not well-supported by experimental evidence. The manuscript is recommended for rejection.

 1 the introduction needs to be rewritten to improve the logical structure. It is hard to find the novelty and significance of this paper.

 2 The ZrO2 coating was used spin-coating method which may not easily transfer to clinic.

 3 The “In vitro biocompatibility evaluation by SBF immersion” didn’t show any bio or cell related experiment.

Author Response

List of the point by point responses to the reviewer’s comments

Reviewer 3

Comments and Suggestions for Authors: Yoshiki et al. reported ZrO2-coated Ti implants with the same topography as that of roughened Ti substrates using the molecular precursor method and study the Influence of the surface chemistry differences for osseointegration. The presentation and organization of the manuscript are unsatisfactory and the conclusions are not well-supported by experimental evidence. The manuscript is recommended for rejection.

1)  the introduction needs to be rewritten to improve the logical structure. It is hard to find the novelty and significance of this paper.

Answer: We added the next sentences and made some revisions in Introduction section at lines 77-81.

We can make clear the difference in bone responses between Ti and ZrO2 surface. The effectiveness of ZrO2 surface on bone response will be clarified in this study. To the best of our knowledge, this is the first trial to elucidate the effectiveness of surface chemical composition of implant materials on bone formation without considering surface structure and roughness.

2)  The ZrO2 coating was used spin-coating method which may not easily transfer to clinic.

Answer: As you commented that spin-coating to dental implant is no easy, but we already succeeded in spin-coating to tooth-shaped and cylindrical implant as shown in the following references.

We added the next sentences and new references at lines 397-406.

In these experiments, we inserted the rectangular plates into the cortical limb bone of rats. This is the basic model for evaluating the bone response to surface-modified materials as previously reported [28-30]. In dental clinics, cylindrical or screw-type implants were normally used. We already preformed the thin film coating to tooth-shaped or cylindrical implants in molecular precursor method using spin-coating technique [37-38]. Moreover, bone responses after the implantation into the jaw bone should be examined. We also developed mandibular bone defect model in rat [15]. As a next series of our experiments, ZrO2 coating was deposited on the cylindrical or screw-type implants and assed the bone formation after the implantation of ZrO2 coating implants into the bone defects in maxilla or mandible.

3) The “In vitro biocompatibility evaluation by SBF immersion” didn’t show any bio or cell related experiment.

Answer: In reference No.34, Kokubo et al. reviewed the history of SBF, correlation of the ability of apatite to form on various materials in SBF with their in vivo bone bioactivities, and some examples of the development of novel bioactive materials based on apatite formation in SBF and concluded that examination of apatite formation on a material in SBF is useful for predicting the in vivo bone bioactivity of a material. Thus, we first employed the SBF immersion experiments and then performed the animal experiments. We changed the headline “In vitro biocompatibility evaluation by SBF immersion” to “SBF immersion experiment” to avoid the confusion, and modified the sentences as following at lines 357-365.

Kokubo et al. [34] reviewed the history of SBF, correlation of the ability of apatite to form on various materials in SBF with their in vivo bone bioactivities, and some examples of the development of novel bioactive materials based on apatite formation in SBF and concluded that examination of apatite formation on a material in SBF is useful for predicting the in vivo bone bioactivity of a material. Thus, we first employed the SBF immersion experiments and then performed the animal experiments. As a result, SBF immersion experiments suggested that the surface property of ZrO2 promotes osteogenesis equivalent to or better than that of Ti in the early stages of bone formation. This result corresponded with that observed in the animal experiments.

Reviewer 4 Report

The paper has merits such as an interesting subject, a suitable title and an informative abstract. The demerits are as following :

a)  the references which despite the fact that are suitable for the content are not enough to sustain novel character,

b)  data presentation which is not well organized and having microstructure images  with not visible features, and not labeled with arrows,

c) the S.I units are not used in whole paper and subchapter 

d) The peacks of XRD and FT-IR spectra are not evidenced in fig 10 a and b and their discussion is week 

e) subchapter  Conclusion is not clearly

Author Response

List of the point by point responses to the reviewer’s comments

Reviewer 4

Comments and Suggestions for Authors

  1. a) the references which despite the fact that are suitable for the content are not enough to sustain novel character,

Answer: We added some references regarding the review for surface modification of zirconia implants at line 43-44.

Surface modifications of Y-TZP implant have been reported [12-14], and large-grit and acid etching procedures were also applied to Y-TZP implants [12-15].

New References

[12] Chopra, D.; Jayasree, A.; Guo, T.; Gulati, K.; Ivanovski, S. Advancing dental implants: Bioactive and therapeutic modifications of zirconia. Bioact Mater. 2021, 13, 161-178. 

[13] Kligman, S.; Ren, Z.; Chung, CH.; Perillo, MA.; Chang, YC.; Koo, H.; Zheng, Z.; Li, C. The Impact of Dental Implant Surface Modifications on Osseointegration and Biofilm Formation. J Clin Med. 2021, 10, 1641.

[14] Schünemann, FH.; Galárraga-Vinueza, ME.; Magini, R.; Fredel, M.; Silva, F.; Souza, JCM.; Zhang, Y.; Henriques, B. Zirconia surface modifications for implant dentistry. Mater Sci Eng C Mater Biol Appl. 2019, 98, 1294-1305. 

  1. b) data presentation which is not well organized and having microstructure images with not visible features, and not labeled with arrows,

Answer: Regarding Figure 4 (new number), we changed the sentence as following at line 223-225 and added arrows.

We added the SEM pictures with higher magnification for PBS and SBF (Figure 10 and 11). 

Both surface showed micro-scale dimples (arrows) on the roughened surface, which were formed by irregular roughness.

  1. c) the S.I units are not used in whole paper and subchapter

Answer: We changed to “Pa” to “N/mm2” at line 105. Will you suggest whether I should change the units elsewhere?

  1. d) The peaks of XRD and FT-IR spectra are not evidenced in fig 10 a and b (Figure 12: new figure number) and their discussion is week

Answer: The broad peaks in XRD and FR-IR is due to the low crystallinity of precipitated apatite. We added the next sentence in Discussion section at line 349.

The broad peaks in XRD and FR-IR is due to the low crystallinity of precipitated apatite.

  1. e) subchapter Conclusion is not clearly

Answer:We added the next sentence in Conclusion at line 423-425.

We can make clear the effectiveness of surface chemical composition of ZrO2 surface on bone formation without considering the factors of surface structure and roughness.

Reviewer 5 Report

Dear authors,

thank you for your study. Overall, major revisions are necessary. Please respond to the following statements.

Headline: Please provide a more concise headline. Please do not use the term "surface chemistry differences" Overall, your study observed surface characteristics and osseointegration of of Ti-implants coated with zirconia. The results were compared to Ti-Implants that were not covered by zirconia.

Abstract: Need to be reworked entirely. Please correct the abstract in accordance to the following points. The introduction and aim of the study must be described more clearly. Information provided in the abstract need to be adopted to the design and results of the study. Methides are missing as well as results that were obtained.

Introduction: Please compress information in the introduction. Information is presented way to extensive. Please focus on the most important issues. Move some of the information to the discussion. Please do not generate redundancy between the introduction and discussion section!

Animal experiment: Describe design of the inserted implants.

line 389: use "evaluated" instead of "demonstrated"

Overall: Find a different definition for the term: “surface chemistry differences" throughout the entire study. Eg. Surface morphology, surface characteristics ect.

Author Response

List of the point by point responses to the reviewer’s comments

Reviewer 5

Comments and Suggestions for Authors: thank you for your study. Overall, major revisions are necessary. Please respond to the following statements.

1) Headline: Please provide a more concise headline. Please do not use the term "surface chemistry differences" Overall, your study observed surface characteristics and osseointegration of of Ti-implants coated with zirconia. The results were compared to Ti-Implants that were not covered by zirconia.

Answer: We changed more concise headline, and changed the term “surface chemistry” to “surface chemical composition” in the whole text including title.

2) Abstract: Need to be reworked entirely. Please correct the abstract in accordance to the following points. The introduction and aim of the study must be described more clearly. Information provided in the abstract need to be adopted to the design and results of the study. Methides are missing as well as results that were obtained.

Answer: We modified the abstract entirely, and added the quantitative results in abstract section. But the abstract should be a total of about 200 words maximum. We added the information as possible as we can in about 200 words.

3) Introduction: Please compress information in the introduction. Information is presented way to extensive. Please focus on the most important issues. Move some of the information to the discussion. Please do not generate redundancy between the introduction and discussion section!

Answer: We compress information in the introduction section.

4) Animal experiment: Describe design of the inserted implants.

Answer: We used the rectangular plate for animal experiment. We modified the sentences about the samples as following at lines 84-92.

Two shapes of Ti samples (JIS2 type, 99.9% mass, Furuuchi Chemical Corp., Tokyo, Japan), disks and rectangular plates were used in this study as shown in Figure 1. Ti disks (12.0 mm in diameter and 1.0 mm in thickness) were used for surface observation, crystal structure and atom elements analyses, phosphate-buffered saline (PBS) and simulated body fluid (SBF) immersion experiments. Ti rectangular plates with dimension of 1.0 mm × 10.0 mm × 20.0 mm were used for zeta potential measurements and those with dimension of 2.0 mm × 3.0 mm × 1.0 mm were for animal implantation experiments. The number of disks and rectangular plates were three for each experiment besides AFM measurements. (n=4)

5) line 389: use "evaluated" instead of "demonstrated"

Answer: We changed the “demonstrated” to “evaluated” according to your comments.

6) Overall: Find a different definition for the term: “surface chemistry differences" throughout the entire study. Eg. Surface morphology, surface characteristics ect.

Answer: We changed the “surface chemistry” to “surface chemical composition”, and “surface topography” to “surface structure and roughness”. We also checked other words according to your comments.

Reviewer 6 Report

Dear Authors, 

you made a really great work!

Author Response

List of the point by point responses to the reviewer’s comments

Reviewer 6

Comments and Suggestions for Authors

Dear Authors,

・The authors are asked to order the abstract according to the division into sections suggested by Mdpi.

Answer: Thank you for reviewing the paper. In Nanomaterials, abstract of other articles are also not divided into sections. We added some information and revised the abstract.

・In the results I ask the authors to explain in a more discursive way the results obtained from the "In vitro biocompatibility evaluation (SBF immersion)" test which I consider extremely useful for the study.

Answer: In reference No.34, Kokubo et al. reviewed the history of SBF, correlation of the ability of apatite to form on various materials in SBF with their in vivo bone bioactivities, and some examples of the development of novel bioactive materials based on apatite formation in SBF and concluded that examination of apatite formation on a material in SBF is useful for predicting the in vivo bone bioactivity of a material. Thus, we first employed the SBF immersion experiments and then performed the animal experiments. We changed the headline “In vitro biocompatibility evaluation by SBF immersion” to “SBF immersion experiment” to avoid the confusion, and modified the sentences as following at lines 357-365.

Kokubo et al. [34] reviewed the history of SBF, correlation of the ability of apatite to form on various materials in SBF with their in vivo bone bioactivities, and some examples of the development of novel bioactive materials based on apatite formation in SBF and concluded that examination of apatite formation on a material in SBF is useful for predicting the in vivo bone bioactivity of a material. Thus, we first employed the SBF immersion experiments and then performed the animal experiments. As a result, SBF immersion experiments suggested that the surface property of ZrO2 promotes osteogenesis equivalent to or better than that of Ti in the early stages of bone formation. This result corresponded with that observed in the animal experiments.

Round 2

Reviewer 3 Report

The manuscript still need to be improved.

Reviewer 4 Report

The authors completed the manuscript and the revised paper is significantly improved

Reviewer 5 Report

Dear authors,

thank you for your corrections. Your study is now suitable.